# High Efficiency of Low Dose Preparations of an Inactivated Lumpy Skin Disease Virus Vaccine Candidate

**DOI:** 10.3390/vaccines10071029

**Published:** 2022-06-27

**Authors:** Janika Wolff, Martin Beer, Bernd Hoffmann

**Affiliations:** Institute of Diagnostic Virology, Friedrich-Loeffler-Institut, Federal Research Institute for Animal Health, Südufer 10, D-17493 Greifswald-Insel Riems, Germany; janika.wolff@fli.de (J.W.); martin.beer@fli.de (M.B.)

**Keywords:** capripox, lumpy skin disease virus, LSDV, inactivated vaccine, minimum protective dose

## Abstract

Capripox virus-induced diseases are commonly described as the most serious poxvirus diseases of production animals, as they have a significant impact on national and global economies. Therefore, they are classified as notifiable diseases under the guidelines of the World Organization for Animal Health (OIE). Controlling lumpy skin disease viral infections is based on early detection, slaughter of affected herds, and ring vaccinations. Until now, only live attenuated vaccines have been commercially available, which often induce adverse effects in vaccinated animals. Furthermore, their application leads to the loss of the “disease-free” status of the respective country. For these reasons, inactivated vaccines have increasingly generated interest. Since 2016, experimental studies have been published showing the high efficacy of inactivated capripox virus vaccines. In the present study, we examined the minimum protective dose of a BEI-inactivated LSDV-Serbia field strain adjuvanted with a low-molecular-weight copolymer adjuvant. Unexpectedly, even the lowest dose tested, with a virus titer of 10^4^ CCID50 before inactivation, was able to provide complete clinical protection in all vaccinated cattle. Moreover, none of the vaccinated cattle showed viremia or viral shedding, indicating the high efficacy of the prototype vaccine even with a relatively low antigen amount.

## 1. Introduction

Lumpy skin disease virus (LSDV), together with sheeppox virus (SPPV) and goatpox virus (GTPV), belongs to the genus *Capripoxvirus* of the family Poxviridae [1]. Capripox viruses in general are described as the most serious poxvirus diseases of domestic and production animals [2,3]. LSDV, which naturally infects cattle and water buffalo [4,5], is mainly transmitted mechanically via blood-feeding insects [6,7,8,9] and possibly via hard ticks [10,11,12]. The clinical course of lumpy skin disease (LSD) ranges from subclinical through mild to acute [13]. After an incubation period of 4–14 days after experimental infection [13,14,15,16] and 1–4 weeks in natural outbreaks [17,18], affected animals develop a fever [15,16,19] and enlarged lymph nodes [16,19], followed by emaciation [20,21], excessive salivation, and nasal discharge [14,16,18]. In addition, characteristic skin lesions appear that are either found sporadically on less hairy regions of the body or generalize and cover the whole body [14,15,16,19]. The morbidity rate is reported to be highly variable, ranging from 3 to 85% [22,23], with an average of approximately 10% [24]. The mortality rate is usually low, at 1 to 3% [18], but can sometimes reach more than 75% [24]. Due to the severe economic impact based on reduced milk production [19,25] and growth rate [20,21], severe hide damage [19,25], expensive control and eradication measures [18,26], and trade restrictions [18], all three capripox virus species are classified as notifiable diseases under the guidelines of the World Organization for Animal Health (OIE) [27]. Controlling capripox viruses is based on early detection of outbreaks, partial or total stamping out, movement restrictions, and vaccination campaigns [26,28]. 

Until now, only live attenuated vaccines have been commercially available [21], which are attenuated via multiple passages in the chorioallantoic membrane of embryonated chicken eggs [29] and in cell cultures [30]. In the field, homologous LSDV-based vaccines [31,32] and heterologous vaccines based on SPPV [33,34,35] or GTPV [31,34,35] are used. However, the administration of these live attenuated vaccines is not authorized in the European Union and not recommended in countries previously free of the disease, since their use compromises the “capripox virus-free” status of the respective country [18,21]. In addition, adverse effects, ranging from increased body temperature [36], through local reactions at the inoculation sites [36,37], to clinical signs similar to clinical LSD [33,38], have been described. Furthermore, vaccine virus DNA has been found in blood samples [33,37], nasal swab samples [39], and skin samples [33,40], and the virus has been successfully isolated from skin nodules [39]. Nevertheless, the transmission of virus strains between vaccinated and naïve animals was not observed [38]. Inactivated vaccines would provide an alternative for the control of capripox viruses, as they are non-replicating and therefore safe and more stable, thus could be used in disease-free countries without loss of this status, and reversion to virulence or transmission could be excluded [41]. Unfortunately, inactivated poxvirus vaccines are reported to be not feasible due to insufficient immunogenicity and short-lived protection [42]. Nevertheless, some successful attempts have been made with inactivated vaccine prototypes against SPPV [43,44] and GTPV [45,46] in sheep and goats, respectively. Recently, an inactivated vaccine prototype against LSDV in cattle was successfully tested by Hamdi et al. [41]. Our recently published results also show good clinical protection and suggest sterile immunity of a Polygen-adjuvanted inactivated vaccine candidate in cattle [47]. Both inactivated prototype vaccines, the one used by Hamdi et al. and ours, were able to induce complete clinical protection in cattle after a severe challenge infection with highly virulent LSDV field strains [41,47]. 

In the present study, the impact of the antigen dose of our Polygen-adjuvanted inactivated vaccine candidate in cattle was examined regarding its protective efficacy. Unexpectedly, a vaccine virus titer of approximately 10^4^ cell culture infectious dose_50_ (CCID_50_)/mL before inactivation (=lowest dose tested in the present study) was sufficient to induce complete clinical protection against a strong challenge with the highly virulent strain already described as LSDV-Macedonia2016. In addition, the antigen concentration comprising a virus titer of 10^4^ CCID_50_/mL before inactivation was able to prevent viremia and viral shedding after challenge infection.

## 2. Materials and Methods

### 2.1. Animals

Twenty-three 3- to 5-month-old female Holstein-Friesian cattle were housed in the facilities of the Friedrich-Loeffler-Institut—Insel Riems, Germany, under biosafety level 4 (animal) conditions. All animals were in good health, and no symptoms indicating acute infections with other pathogens could be observed. All respective animal protocols were reviewed by a state ethics commission and were approved by the competent authority (State Office for Agriculture, Food Safety and Fisheries of Mecklenburg-Vorpommern, Rostock, Germany; Ref. No. 7221.3-1-002/20; approval date 31 March 2020).

### 2.2. Vaccine Preparation

An inactivated antigen based on the LSDV-Serbia field strain [47] was prepared by Zoetis (Olot, Spain) and tested according to inactivation and sterility. The titer of antigen stock before inactivation was 10^6^ cell culture infectious dose_50_ (CCID_50_)/mL on the BHK-21 cell line. The inactivated LSDV antigen preparation was then sent to the Friedrich-Loeffler-Institut on dry ice. There, the antigen dilutions were prepared, and the adjuvant was added. The antigen was used undiluted or diluted 1:10 (calculated titer before inactivation 10^5^ CCID_50_/mL on BHK-21 cell line) and 1:100 (calculated titer before inactivation 10^4^ CCID_50_/mL on BHK-21 cell line) in TE buffer at pH 8.0, and 10% Polygen (lot no. P10061; MVP Adjuvants^®^, Omaha, NE, USA) was added directly before the first immunization. Using real-time qPCR, the viral genome load of all three vaccine preparations was defined semi-quantitatively. For the undiluted vaccine preparation, Cq values of 14.2 and 15.0 before and after adding the adjuvant were ascertained. Accordingly, for the 1:10 diluted vaccine, Cq values of 17.2 and 18.4, and for the 1:100 diluted vaccine, Cq values of 20.5 and 20.9 were determined.

The TE buffer used for dilution of the antigen and PBS administered to the challenge control group are commercially available and endotoxin-free. 

Prepared prototype vaccines were stored at 4 °C in the dark until secondary immunization. Each animal received 2 mL of the prototype at first and secondary immunizations, since this vaccination scheme proved successful in our recently performed proof-of-concept study [47]. 

### 2.3. Experimental Design and Sample Collection

Cattle in group A were vaccinated with the undiluted antigen vaccine, cattle in group B received the antigen diluted 1:10, and cattle in group C were immunized with the antigen diluted 1:100. Cattle in group D served as the control group and were inoculated with PBS. All animals were inoculated intramuscularly in the neck with 2 mL of the respective vaccine. Primary immunization was performed at day 0 of the animal trial, followed by secondary immunization 21 days post (primary) vaccination (dpv). Challenge infection was performed intravenously with 2 mL of the highly virulent LSDV-Macedonia2016 field strain 42 days post primary immunization (42 dpv ≅ 0 days post challenge (dpc)). Back-titration of the challenge virus on MDBK cells revealed a titer of 10^6·6^ CCID_50_/mL.

Body temperature was measured daily from −4 dpv until 28 dpc, and an increasing temperature higher than 40 °C was defined as fever. A clinical reaction score (CRS) evaluating general condition, feed and water intake, respiratory signs, skin lesions, and expansion of lymph nodes [16] was calculated from 0 to 14 dpc, and human end point was defined as CRS ≥ 10. Since all animals in the challenge control group displayed lumpy skin disease (LSD) at this time point, and neither increased body temperature nor clinical signs could be observed in all three vaccinated groups, a detailed examination of the CRS was stopped at 14 dpc.

Different samples were taken at defined time points during the animal trial, using EDTA blood samples for cell-associated viremia, serum samples for analysis of cell-free viremia, and nasal swab samples for determination of viral shedding. In addition, serum samples were used to examine seroconversion. Sampling was performed at 0 and 21 dpv, then 0, 3, 5, 7, 10, 12, 14, 21, and 28 dpc. During necropsy, the following organ samples were taken: cervical lymph node, mediastinal lymph node, liver, and lung. Samples of organs and skin areas displaying characteristic pox-like lesions were also taken.

### 2.4. Molecular Diagnostics

Organ samples were homogenized in a serum-free medium using the TissueLyser II tissue homogenizer (Qiagen, Hilden, Germany). The extraction of homogenized tissue samples and samples taken during the animal trial was performed using the NucleoMag Vet kit (Macherey-Nagel, Düren, Germany) according to the manufacturer’s instructions, with volume modifications as described before [48], utilizing the KingFisher Flex System (Thermo Scientific, Darmstadt, Germany). During the extraction process, an internal control (IC-2 DNA) was added for control of successful DNA extraction and inhibition-free amplification [49]. For an analysis of the capripox virus genome, the pan capripox real-time qPCR described by Bowden et al. [50] with a modified probe of Dietze et al. [51] was performed utilizing the PerfeCTa qPCR ToughMix (Quanta BioSciences, Gaithersburg, MD, USA).

### 2.5. Serological Examination

Two methods were used for the serological analysis. For the LSDV-specific serum neutralization assay (SNT), serum samples were heat-inactivated for 30 min at 56 °C. Then, the log2 dilution series in serum-free medium were prepared in triplicate starting from 1:10, using a 96-well plate format. Subsequently, 50 µL of LSDV-Neethling vaccine strain [14] with a titer of 10^3·3^ CCID_50_/mL was added to each well. Mixtures of serum dilutions and virus were incubated for 2 h at 37 °C and 5% CO_2_. Afterwards, 100 µL of MDBK cells (approximately 30,000 cells) was added to each well, followed by incubation for 7 days at 37 °C and 5% CO_2_. The development of CPE was analyzed at 7 days post infection using a Nikon Eclipse TS-100 light microscope. To determine the neutralizing titer, the method of Spearman and Kärber was applied [52,53]. Samples with a neutralizing titer similar to or higher than 1:13 were defined as positive.

Along with the SNT, the ID Screen^®^ Capripox Double Antigen (DA) ELISA from ID.vet (Montpellier, France) was performed according to the manufacturer’s instructions. To define the ELISA titer, log2 dilution series of the collected serum in PBS were performed and analyzed, and the titer was defined as the highest dilution still positive in the ELISA.

## 3. Results

### 3.1. Safety Observation and Clinical Scoring

After primary and secondary immunization, no adverse effects could be observed. All animals behaved normally, and no swelling at the inoculation site could be detected. Some animals displayed increased body temperature or fever for single days, mainly after secondary vaccination, and especially in the groups that were immunized with the diluted antigen (Figure 1A–C). However, as this phenomenon was also seen to a certain extent in the challenge control group inoculated with PBS (Figure 1D), a direct correlation with the vaccines can be neither confirmed nor rejected.

After the challenge infection, the body temperature of all cattle in the vaccination groups remained in the normal range, and no marked differences could be observed between days post vaccination and post challenge (Figure 1A–C). In contrast, the body temperature of three out of five animals in the challenge control group started to increase between 3 dpc (R/728) and 6 dpc (R/762, R/988). While these three animals developed a severe clinical course of lumpy skin disease (LSD) and displayed fever up to 40.8 °C (R/728, 4 dpc; R/988, 9 dpc) and 40.5 (R/762, 11 dpc), the body temperature of the two remaining challenge control animals (R/714, R/648) remained unremarkable (Figure 1D).

The clinical reaction score was determined daily from 0 until 14 dpc. All animals inoculated with the different vaccine prototypes were completely protected against challenge infection by the highly virulent LSDV-Macedonia2016 field strain, independent of the antigen amount (Figure 2A–C). Only R/643 and R/760 in group C (antigen diluted 1:100) displayed slightly increased cervical lymph nodes for 2 days (10 and 11 dpc) and 1 day (10 dpc), respectively (Figure 2C). On the contrary, animals in the challenge control group showed typical clinical courses of LSDV infection. Here, clinical signs started at 4 dpc (R/728) and 5 dpc (R/714, R/684, R/762, R/988). One animal showed only a very mild course, with a maximum clinical score of 1 (R/684), due to slightly reduced food intake and a slightly enlarged cervical lymph node on the side of the challenge virus inoculation. R/714 developed a moderate course of LSDV infection with a maximum clinical reaction score of 4.5 at 8 dpc. The remaining three cattle of this group were severely affected by the challenge infection and reached the human end point at 10 dpc (R/728, CRS = 10; R/988, CRS = 13) and 11 dpc (R/762, CRS = 10) (Figure 2D).

### 3.2. Replication and Shedding of Virus

For the analysis of viremia, EDTA blood samples for cell-associated viremia and serum samples for cell-free viremia were tested using the pan capripox real-time qPCR assay [50] at defined time points after the challenge infection. Interestingly, viremia could not be observed in any of the three vaccination groups at all time points (Figure 3A–C). Solely for R-759, EDTA blood samples tested positive for viral genome once at 12 dpc with a comparably high Cq value (33.6) (Figure 3B). 

In contrast, viremia could be observed in the EDTA blood and serum samples of all cattle in the challenge control group. Here, viremia was first detected in the EDTA blood samples of three out of five animals at 5 dpc (R-728, R-762, R-988) and in the serum of R-762, with Cq values between 33 and 36. These three animals had to be euthanized due to ethical reasons at 10 dpc (R-728, R-988) and 11 dpc (R-762), at which time they displayed both cell-associated and cell-free viremia and had Cq values between 28.8 (R-762, EDTA blood sample) and 34.7 (R-728, serum). Viremia of R-714 in group D started at 10 dpc and lasted until 14 dpc (Cq values between 31.2 and 38.3). For R-684, viremia could only be observed at 12 dpc in the EDTA blood samples, with a very high Cq value of 38.5 (Figure 3D). 

Viral shedding was analyzed using nasal swab samples taken at defined days post challenge infection. Like viremia, the shedding of viral DNA could not be observed in any animals previously inoculated with the inactivated vaccine independent of the antigen concentration and without exception (Figure 4A–C).

On the contrary, viral shedding could be observed in the three animals in the challenge control group that were euthanized before the end of the study due to severe LSD. First, shedding of viral DNA was detected in the nasal swab samples of R-728 and R-988 at 7 dpc, followed by R-762 at 10 dpc. Whereas Cq values were comparably high at 7 dpc (around 37–38), all three cattle displayed strong nasal shedding with Cq values between 23.6 (R-988) and 28.1 (R-728) at 10 dpc. Nasal swab samples of both surviving animals in the challenge control group remained negative during the whole study (Figure 4D).

### 3.3. Viral Genome Load in Different Organ Samples

During necropsy, cervical and mediastinal lymph node samples, liver, and lung were taken and analyzed for their viral genome load. All organ samples taken from vaccinated cattle tested negative for the capripox virus genome. The same result could be observed for both surviving animals of group D (R-714, R-684). In contrast, both lymph nodes and lungs of all three cattle euthanized due to a severe clinical course scored positive on the pan capripox real-time qPCR, displaying a generalized LSDV infection. The highest viral genome load could be detected in the cervical lymph node and lung (Cq values of 24.8 to 32.3). However, the liver was negative for all animals in all groups (Appendix A).

In addition to the defined organ panel, skin and lung lesion samples were taken from the three severely affected cattle in the challenge control group during necropsy. All lung lesions were highly positive for the capripox virus genome, with Cq values between 20.9 (R-762) and 30.6 (R-728). For skin samples taken from nodules located in the neck and udder region of R-728 and R-988, Cq values ranged from 16.9 (R-988, udder) to around 20 (R-728, R-988) and up to 26.9 (R-988, neck).

### 3.4. Serological Examination

The serological response was examined using DA ELISA and SNT. Before the beginning of the study, all animals were negative for capripox-specific antibodies, which were shown by both tests (Table 1).

In all three vaccinated groups, five out of six animals had positive ELISA scores on the day of challenge infection. The last animal in each group (group A, R/765; group B, R/729; group C, R/760) showed a positive ELISA result at 7 dpc. However, these three animals with a negative result also showed some reactivity on ELISA below the cut-off of 30 S/P% at the time of challenge. A similar pattern can be seen for the SNT results. Here, all cattle in the vaccination groups showed positive results on the SNT on the day of the challenge infection. It is notable that three out of six animals in group C (antigen diluted 1:100) showed a very weak positive SNT result (1:13) at this time. However, all cattle, independent of the vaccination group, showed strong seroconversion on ELISA and SNT at least at 7 dpc (Table 1). The increased seroconversion after challenge over that time was confirmed by analyzing the ELISA titer beside the SNT titer (Table 2). The result let us suppose that strong LSDV-specific antibody production was initiated immediately after the challenge infection, without any indication of detectable virus replication in the PCR-tested matrices. Furthermore, no general difference in the level of serological titers was found between vaccination groups, indicating a similar immune response (Table 2).

In contrast, none of the three animals in the challenge control group that had to be euthanized at 10 and 11 dpc due to a severe clinical course showed a serological response on either test, which was due to the early time point of euthanasia. Both surviving cattle in group D scored positive for antibodies on ELISA and SNT at 21 or 28 dpc (R/714: 21 dpc, ELISA and SNT; R/684: 21 dpc, ELISA; R/684: 28 dpc, SNT) (Table 1).

## 4. Discussion

Inactivated vaccines provide several advantages compared to live attenuated vaccines, as they are non-replicating and therefore safe. Problems with contaminating pathogens or recombinants with virulent field viruses are avoided. In addition, they are more stable than most live attenuated vaccines [41,44]. Until now, only live attenuated vaccines have been commercially available to control capripox viruses [21]. Despite the common knowledge that poxvirus vaccines require a replicating antigen for the induction of sufficient and long-lived protection [42,54], recent studies suggest that inactivated capripox virus vaccines are a good and efficient alternative [41,44,47]. Especially in countries with advanced cattle breeding and production, the prophylactic application of inactivated LSDV vaccines without any side effects should be of interest. During the present study, we examined the dose dependency of our recently developed BEI-inactivated capripox vaccine based on an LSDV field strain propagated in non-bovine cells and adjuvanted with a low-molecular-weight copolymer [47]. 

As seen in our previous study [47], the vaccine prototype did not induce any local adverse effects or fever in the vaccinated animals and therefore can be considered safe for administration in cattle (Figure 1A–C). In addition, no clinical signs could be observed in any of the three vaccinated groups independent of the antigen amount (Figure 2A–C). The only exceptions were R/643 and R/760 in group C (antigen diluted 1:100), which showed slightly increased cervical lymph nodes for a maximum of two days (Figure 2C); this might have been a very slight reaction to the challenge virus. However, neither animal showed viremia or nasal shedding of the virus after the challenge infection. In contrast to the vaccinated groups, all five animals in the challenge control group became infected after being inoculated with the challenge material. Here, two animals displayed a mild and moderate course of LSD, and the remaining three cattle developed a severe clinical course and had to be euthanized for animal welfare reasons before the end of the study (Figure 2D). Thus, we can conclude that the challenge model worked as robustly as in previous experimental infection studies with LSDV-Macedonia2016 [14,16,47]. In summary, these data indicate that all three antigen amounts tested were able to induce complete clinical protection in cattle. 

In addition, molecular data also show successful protection of all animals vaccinated with one of the three antigen amounts. Specifically, all antigen concentrations used in the different vaccine batches were able to completely prevent viremia (Figure 3A–C). Solely in the EDTA blood sample of the R-759 could the viral genome be detected, and at a single time point (Figure 3B). However, considering all the other EDTA blood and serum samples taken from the vaccinated animals, including R-759, independent of the antigen concentration, and the group that received the vaccine with a lower antigen concentration than R-759 (Figure 3A–C), this has to be interpreted with caution. Contamination during the sampling procedure or sample handling cannot be fully excluded. On the other hand, local replication, e.g., in lymph nodes, and timely limited spreading of the virus in the blood system at a very low level should also not be excluded. On the contrary, all animals in the challenge control group showed viremia, and in the three animals that developed severe LSD, the viral genome could be observed in the nasal swab samples (Figure 3D and Figure 4D). 

Whereas in our previous study, all animals vaccinated with this vaccine prototype had positive DA ELISA scores on the day of the challenge infection [47], in this study, animals in all three vaccination groups had positive SNT scores and showed at least serological reactivity on ELISA at 0 dpc. Thus, a serological response was at least detectable on the day of the challenge infection in all vaccinated animals (Table 1). A comparison of overall titers between groups did not show any marked difference (Table 2), indicating a similar immune response to the vaccine in each group. Nevertheless, the serum neutralizing titer of three animals in group C (R-703, R-727, R-760) (1:100 antigen dilution) was very low, with an SNT titer of 1:13, compared to the SNT titer of the other vaccine groups (A, B) on the day of the challenge.

Taking all the results together, they support the clinical findings and point to complete protection against LSDV infection induced by the inactivated vaccine prototype used in this study. This is in line with our previous experience with a BEI-inactivated LSDV field strain formulated with the adjuvant Polygen [47], and with the findings of Hamdi et al. who used an inactivated LSDV Neethling vaccine strain in cattle [41], and Boumart et al. who successfully tested an inactivated vaccine against SPPV in sheep [44]. Hamdi et al. used a virus titer before inactivation of approximately 10^6^ TCID_50_/animal [41], which is similar to our group A, which received the undiluted antigen. In the study of Boumart et al. a lower vaccine virus stock with a titer of 10^5.5^ TCID_50_/mL was used for vaccine preparation [44]. This made us hopeful that even a lower titer of our inactivated prototype vaccine than used before (10^7^ CCID_50_/animal in the previous study [47]) would be able to induce sufficient protection against severe LSDV challenge infection in cattle. Therefore, we tested vaccine virus titers before the inactivation of 10^6^, 10^5^, and 10^4^ CCID_50_/animal. Unexpectedly, no differences in protection efficacy could be observed between the three groups in the present study (Figure 1, Figure 2 and Figure 3), indicating that our prototype vaccine still has suitable efficacy with a relatively low antigen concentration. These findings strengthen the possibility of commercial production of the inactivated vaccine candidate, since this titer can be easily achieved in the used non-bovine cell line, and no costly purification or concentration steps are needed. An evaluation of the cellular immune response analyzing interferon γ or interleukin 4 production and lymphocyte proliferation [34,35,36,55] could give further insight into the mechanisms of protection of our vaccine prototype and point out differences between different antigen amounts. In addition, an actual minimum protective dose as well as the duration of immunity after vaccination should be addressed in future studies to provide additional information about the protection efficacy of inactivated LSDV prototype vaccines in general and over time. 

## 5. Conclusions

In conclusion, the results obtained during the present study strongly support the possibility of controlling capripox virus-induced diseases with inactivated vaccines. All animals vaccinated with a BEI-inactivated and Polygen-adjuvanted LSDV field strain were completely protected against clinical signs, viremia, and viral shedding after a strong challenge infection with the highly virulent LSDV-Macedonia2016 field strain independent of the antigen amount used. Even an antigen concentration equivalent to a vaccine virus titer of 10^4^ CCID_50_/animal before inactivation was able to induce sufficient protection in vaccinated cattle. Further studies are necessary to evaluate the quality and efficacy of the immunological response to the inactivated vaccine several months after vaccination.

## Figures and Tables

**Figure 1 vaccines-10-01029-f001:**
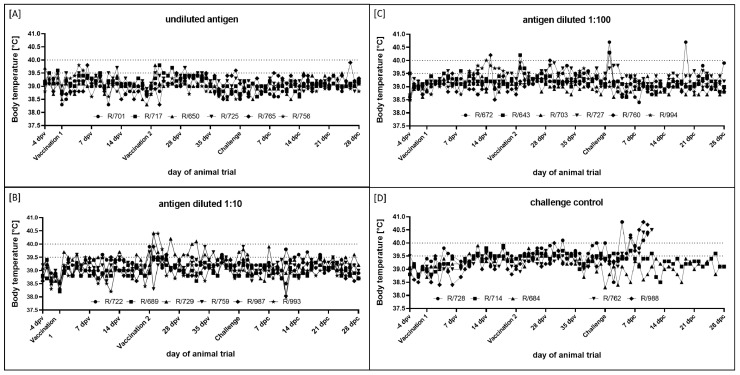
Body temperature of cattle immunized with inactivated LSDV vaccine candidates with different amounts of antigen. Body temperature was measured daily from −4 dpv until 28 dpc. Animals were inoculated with inactivated LSDV vaccine candidate with different amounts of antigen and then challenged with virulent LSDV-Macedonia2016. (**A**) Vaccine virus titer before inactivation was 10^6^ CCID_50_/mL on BHK-21 cell line. (**B**,**C**) Antigen was diluted 1:10 and 1:100, respectively, before addition of adjuvant. (**D**) Cattle in group D were housed as challenge control group and received PBS instead of vaccine. Body temperature >39.5 °C was defined as increased, and body temperature >40.0 °C indicated fever.

**Figure 2 vaccines-10-01029-f002:**
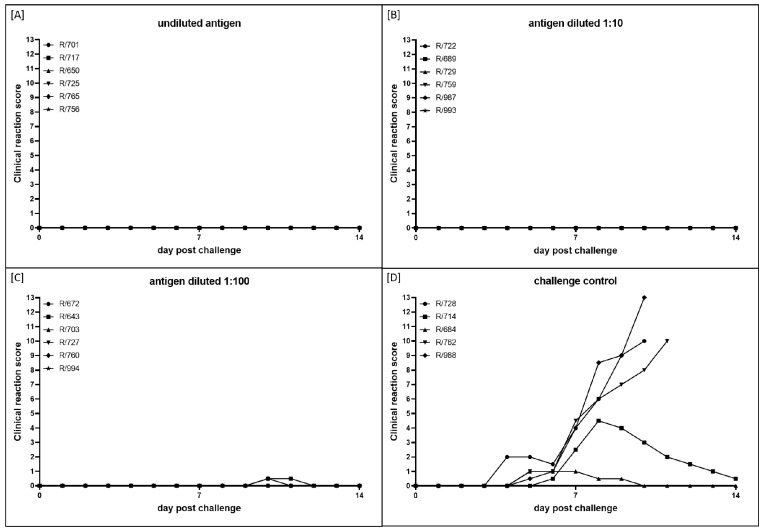
Clinical reaction scores of cattle immunized with inactivated LSDV vaccine candidates with different amounts of antigen. Clinical reaction score was measured daily from 0 until 14 dpc. Animals were inoculated with inactivated LSDV vaccine candidate with different amounts of antigen and then challenged with virulent LSDV-Macedonia2016. (**A**) Vaccine virus titer before inactivation was 10^6^ CCID_50_/mL on BHK-21 cell line. (**B**,**C**) Antigen was diluted 1:10 and 1:100, respectively, before adjuvant was added. (**D**) Cattle in group D were housed as challenge control group and received PBS instead of vaccine.

**Figure 3 vaccines-10-01029-f003:**
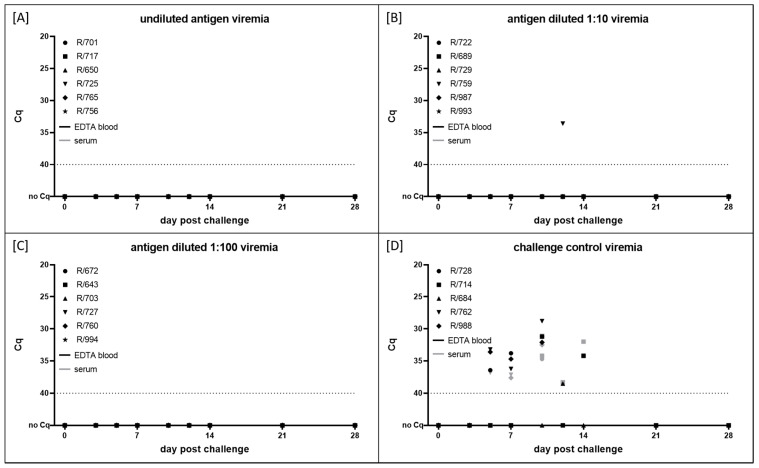
Cell-associated (EDTA blood samples, black marks) and cell-free viremia (serum, gray marks) after challenge infection. Animals were vaccinated with inactivated LSDV vaccine candidate with different amounts of antigen and then challenged with virulent LSDV-Macedonia2016. (**A**) Vaccine virus titer before inactivation was 10^6^ CCID_50_/mL on BHK-21 cell line. (**B**,**C**) Antigen was diluted 1:10 and 1:100, respectively, before adjuvant was added. (**D**) Cattle in group D were housed as challenge control group and received PBS instead of vaccine. Cut-off was set at Cq 40.0.

**Figure 4 vaccines-10-01029-f004:**
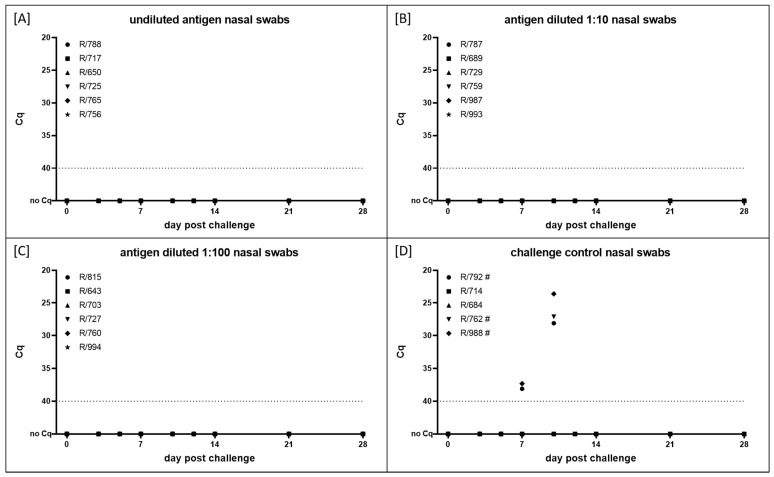
Viral shedding after challenge infection. Animals were inoculated with inactivated LSDV vaccine candidate with different amounts of antigen and then challenged with virulent LSDV-Macedonia2016. (**A**) Vaccine virus titer before inactivation was 10^6^ CCID_50_/mL on BHK-21 cell line. (**B**,**C**) Antigen was diluted 1:10 and 1:100, respectively, before adjuvant was added. (**D**) Cattle in group D were housed as challenge control group and received PBS instead of vaccine. Cut-off was set at Cq 40.0.

**Table 1 vaccines-10-01029-t001:** Serological results of vaccinated and challenge control cattle on ELISA and SNT from −2 dpv until 28 dpc (S/P% below 30 on ELISA and SNT titer < 1:10 was defined as negative; positive values are given in boldface for easier understanding).

	Group A (Antigen Undiluted)		Group B (Antigen Diluted 1:10)		Group C (Antigen Diluted 1:100)		Group D (No Antigen)	
dpv/dpc	DA ELISA S/P%	SNT Titer	DA ELISA S/P%	SNT Titer	DA ELISA S/P%	SNT Titer	DA ELISA S/P%	SNT Titer
	R-701		R-722		R-672		R-728	
2 dpv	0	<1:10	0	<1:10	0	<1:10	−1	<1:10
21 dpv	7	**1:13**	4	<1:10	1	<1:10	−1	<1:10
42 dpv/0 dpc	**60**	**1:25**	**75**	**1:40**	**78**	**1:40**	0	<1:10
7 dpc	**188**	**1:200**	**209**	**1:128**	**110**	**1:200**	0	<1:10
14 dpc	**187**	**1:256**	**197**	**1:80**	**148**	**1:128**	0	<1:10
21 dpc	**178**	**1:100**	**182**	**1:100**	**140**	**1:320**		
28 dpc	**174**	**1:80**	**160**	**1:128**	**128**	**1:256**		
	R-717		R-689		R-643		R-714	
2 dpv	1	<1:10	0	<1:10	3	<1:10	−1	<1:10
21 dpv	2	<1:10	0	<1:10	2	<1:10	0	<1:10
42 dpv/0 dpc	**117**	**1:64**	**50**	**1:64**	**41**	**1:20**	0	<1:10
7 dpc	**187**	**1:80**	**59**	**1:32**	**219**	**1:64**	0	<1:10
14 dpc	**183**	**1:160**	**66**	**1:50**	**253**	**1:160**	15	<1:10
21 dpc	**167**	**1:80**	**82**	**1:32**	**242**	**1:200**	**103**	**1:64**
28 dpc	**152**	**1:50**	**87**	**1:32**	**249**	**1:160**	**134**	**1:80**
	R-650		R-729		R-703		R-684	
2 dpv	0	<1:10	0	<1:10	0	<1:10	−1	<1:10
21 dpv	3	**1:13**	0	<1:10	−1	<1:10	0	<1:10
42 dpv/0 dpc	**42**	**1:32**	24	**1:64**	**50**	**1:13**	0	<1:10
7 dpc	**143**	**1:50**	**99**	**1:256**	**177**	**1:100**	0	<1:10
14 dpc	**116**	**1:100**	**111**	**1:128**	**160**	**1:50**	7	<1:10
21 dpc	**112**	**1:50**	**100**	**1:80**	**145**	**1:50**	**32**	**1:13**
28 dpc	**113**	**1:160**	**95**	**1:64**	**125**	**1:80**	**77**	**1:32**
	R-725		R-759		R-727		R-762	
2 dpv	0	<1:10	0	<1:10	−1	<1:10	0	<1:10
21 dpv	3	<1:10	0	<1:10	0	<1:10	0	<1:10
42 dpv/0 dpc	**68**	**1:16**	**47**	**1:50**	**72**	**1:13**	0	<1:10
7 dpc	**132**	**1:100**	**81**	**1:100**	**170**	**1:32**	0	<1:10
14 dpc	**216**	**1:200**	**159**	**1:800**	**170**	**1:50**	0	<1:10
21 dpc	**223**	**1:800**	**204**	**1:1600**	**167**	**1:50**		
28 dpc	**223**	**1:640**	**203**	**1:1280**	**167**	**1:64**		
	R-765		R-987		R-760		R-988	
2 dpv	0	<1:10	0	<1:10	0	<1:10	0	<1:10
21 dpv	1	<1:10	3	<1:10	0	<1:10	0	<1:10
42 dpv/0 dpc	14	**1:64**	**96**	**1:128**	9	**1:13**	0	<1:10
7 dpc	**83**	**1:100**	**208**	**1:160**	**119**	**1:80**	0	<1:10
14 dpc	**238**	**1:512**	**247**	**1:256**	**186**	**1:100**	0	<1:10
21 dpc	**278**	**1:1280**	**245**	**1:512**	**198**	**1:200**		
28 dpc	**271**	**1:1600**	**243**	**1:400**	**187**	**1:200**		
	R-756		R-993		R-994			
2 dpv	0	<1:10	0	<1:10	0	<1:10		
21 dpv	1	<1:10	4	**1:13**	0	<1:10		
42 dpv/0 dpc	**86**	**1:25**	**38**	**1:40**	**42**	**1:40**		
7 dpc	**133**	**1:50**	**126**	**1:32**	**115**	**1:128**		
14 dpc	**166**	**1:50**	**138**	**1:32**	**140**	**1:64**		
21 dpc	**163**	**1:50**	**143**	**1:32**	**127**	**1:80**		
28 dpc	**169**	**1:100**	**140**	**1:50**	**120**	**1:64**		

**Table 2 vaccines-10-01029-t002:** Comparison of development of ELISA and SNT titers after challenge.

Group	Cattle	DA ELISA Titer			SNT Titer		
		0 dpc	7 dpc	28 dpc	0 dpc	7 dpc	28 dpc
Group A: undiluted antigen	R-701	1:2	1:16	1:16	1:25	1:200	1:80
	R-717	1:8	1:32	1:8	1:64	1:80	1:50
	R-650	original	1:16	1:8	1:32	1:50	1:160
	R-725	1:2	1:8	1:64	1:16	1:100	1:640
	R-765	negative	1:4	1:128	1:64	1:100	1:1600
	R-756	1:4	1:8	1:16	1:25	1:50	1:100
Group B: antigen diluted 1:10	R-722	1:8	1:64	1:16	1:40	1:128	1:128
	R-689	1:2	1:2	1:4	1:64	1:32	1:32
	R-729	negative	1:16	1:8	1:64	1:256	1:64
	R-759	1:2	1:4	1:64	1:50	1:100	1:1280
	R-987	1:8	1:32	1:128	1:128	1:160	1:400
	R-993	1:2	1:16	1:16	1:40	1:32	1:50
Group C: antigen diluted 1:100	R-672	1:4	1:16	1:16	1:40	1:200	1:256
	R-643	original	1:64	1:128	1:20	1:64	1:160
	R-703	1:2	1:64	1:32	1:13	1:100	1:80
	R-727	1:4	1:64	1:32	1:13	1:32	1:64
	R-760	negative	1:16	1:64	1:13	1:80	1:200
	R-994	1:2	1:16	1:16	1:40	1:128	1:64

## Data Availability

The data presented in this study are available from the corresponding author upon request. The data are not publicly available due to funding by a third party.

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
