# Peer review of "High Efficiency of Low Dose Preparations of an Inactivated Lumpy Skin Disease Virus Vaccine Candidate"

_vaccines, 2022, doi:10.3390/vaccines10071029_

Round 1

Reviewer 1 Report

The authors purported to uncover a minimal protective dose of inactivated vaccine against LSDV in cattle. They could show that BEI-inactivated vaccine of LSDV could protect against an another LSDV challenge.

Unfortunately, the manuscript is immature and lacks precision: e.g. the minimal dose of 104 CCID50 is not really shown to be the lowest concentration of inactivated LSDV that gives 50% protection against the challenge. To satisfy the title statement one would expect that the authors would start the manuscript with presented data and would continue the challenge experiments with lower concentrations than 104 CCID50.

Another confusion is what type of vaccine did the authors used. Is it the LSDV-Serbia field strain or another inactivated capripox virus? Please, keep it named the same throughout the manuscript: e.g. ”Inactivated LSDV-Serbia field strain” or another name.

Please indicate how many females and males were used in the four experimental groups in M&M. It would be even useful to connect it to the cattle recognition numbers. Since, it is known that females are more resistant against viral infections.

Why are the CRS and viremia data graphics not completed with the two remaining control animals till 28 dpc?

Why is the Cq-axe in Figure 3 reversed?

Table 1, Table 2 and Table 3 should be put in present form in supplementary data.

In Results, I would present a more digested version of the data for the three tables and for the Tables 2 and 3 I would indicate even relative measured numbers.

Reviewer 2 Report

From my point of view, this work is fair and provides interesting results for the study of the minimum protective dose for LSDV inactivated vaccine candidates. However, this study has the following problems:

  1. Although the results show that inactivated vaccines are safe and reliable, I think it is necessary to increase the sterility test of inactivated vaccines and whether the antigen is completely inactivated;
  2. Keywords LSDV and massive dermatosis virus repetition;
  3. Line 180: The units of 10dpi and 11dpc should be unified;
  4. Line 93:  It should be" on unit BHK-21"instead of"on BEK-21".

Reviewer 3 Report

The article deals with the big problem of the large diffusion of lumpy skin disease due to its high morbidity, and the related vaccination generally possible only with live attenuated vaccines that often induce adverse effects in vaccinated animals. The authors propose the use of an inactivated vaccine, concluding that this works independently of the contained antigenic dose.

The idea is good, but there are some criticisms to be clarify and an English revision is needed.

Then, the paper can be published after some corrections/adding/clarifications and after a thorough review of the English language and the correctness of the construction of sentences.

Reviewer 4 Report

  • The title of the article is a bit misleading and not technically accurate. The authors actually did not find the minimum protective dose of their inactivated vaccine. The lowest dose used in the study provided the same level protection as the highest dose.
  • line 91: explain how the inactivation was performed
  • In the figures, I do not like how the antigen dose is depicted. I think labeling the undiluted as 1x10^6 CCID50/mL then 10^5 and 10^4 (instead of 1:10 and 1:00, respectively) is fine.
  • I think "challenge control" is better labeled as "PBS" to make clear to the reader what treatment this group of animals received.
  • line 106: briefly explain why the i.v. route was chosen. Was it to mimic infection by a blood-feeding insect?
  • The Methods should include a description of the criteria used to determine when an animal would be euthanized.
  • The Methods should also describe the criteria for how clinical reaction scores were determined.
  • Figure 1: Is a temp greater than 40 deg C a standard criteria for the determination that a fever is present? Briefly explain this is in the Methods.
  • Figure 1: These figure panels are very hard to examine since there are so many lines. I appreciate that the authors are trying to show data from individual animals, however. Can composite data also be shown (for example, the average temp and std dev for each group at each time point)?
  • Lines 181 and 227: I would suggest replacing "Contrarily" with "In contrast,..."
  • Lines 187-190: This is confusing. Please rephrase for clarity. Also, why is a human endpoint being referenced when this is a study of cattle?
  • Methods: Explain more clearly what is meant by "EDTA blood." This is not a commonly used phrase in my opinion and it's unclear what it means or what was being measured. I would not use this term and would recommend choosing a different label.
  • Table 1 isn't really necessary. The data can be written in text form in the Results section.
  • Table 2 is somewhat overwhelming to make sense of all the numbers. Could the composite or aggregate data conveyed in Table 2 be shown as a Figure in place of or in addition to this table?
  • I thought the Discussion section was well written and thought out.